**communications** engineering

# StarWhisper Telescope: an AI framework for automating end-to-end astronomical observations
Cunshi Wang [1,2] ✉, Yu Zhang [2], Yuyang Li [1,2] ✉, Xinjie Hu[2,3], Yiming Mao[1,2,4], Xunhao Chen [1,2], Pengliang Du[2], Rui Wang[2], Ying Wu[2,5], Hang Yang[1,2], Yansong Li [6], Beichuan Wang[1,2], Haiyang Mu[2], Xiaohan Chen[1,2], Shunxuan He[1,2], Hao Mo[1,2], Liyue Zhang[1,2], Lin Du[1,2], Yunning Zhao[1,2], Jianfeng Tian [2], Liang Ge[2,5], Yongna Mao[1], Shengming Li[2], Zheng Wang[2], Xiaomeng Lu[2], Jinhang Zou[2], Yang Huang [1,2], Ningchen Sun[1,2], Jie Zheng[2], Min He[2], Yu Bai[2,5], Junjie Jin[2], Hong Wu[1,2] & Jifeng Liu [1,2,5]

The exponential growth of large-scale telescope arrays has boosted time-domain astronomy development but introduced operational bottlenecks, including labor-intensive observation planning, data processing, and real-time decision-making. Here we present the StarWhisper Telescope system, an AI agent framework automating end-to-end astronomical observations for surveys like the Nearby Galaxy Supernovae Survey. By integrating large language models with specialized function calls and modular workflows, StarWhisper Telescope autonomously generates site-specific observation lists, executes real-time image analysis via pipelines, and dynamically triggers follow-up proposals upon transient detection. The system reduces human intervention through automated observation planning, telescope controlling and data processing, while enabling seamless collaboration between amateur and professional astronomers. Deployed across Nearby Galaxy Supernovae Survey's network of 10 amateur telescopes, StarWhisper Telescope has detected transients with promising response times relative to existing surveys. Furthermore, StarWhisper Telescope's scalable agent architecture provides a blueprint for future facilities like the Global Open Transient Telescope Array, where AI-driven autonomy will be critical for managing 60 telescopes.

As time domain astronomy develops, large scientific facilities are in the planning. The Global Open Transient Telescope Array (GOTTA[1,2], which also referred to the SiTian project.), aims to construct 60 1-m telescopes in its first stage to build an array to monitor the sky. The labor cost for the observation of the GOTTA is huge, with an estimated observation personnel sequence exceeding 200 people. Most of the telescopes will be deployed at the Lenghu Observatory, with a latitude of about 4200 m. Due to the large number of telescopes, automated observations are necessary for future time-domain surveys. With the help of a Large Language Model (LLM) based agent, new automated observations will aid astronomers by taking the observations and processing the data, realizing the concept of an "AI Astronomer".

Astronomical observation can be broadly divided into three phases: (1) observation planning, (2) observation execution, and (3) data processing. These phases are executed sequentially, starting with planning, followed by execution, and concluding with data processing. Notably, the results of data processing feed back into the planning phase, influencing future observation strategies and helping to optimize the overall observational process.

The observation planning phase aims to create an observation list for each telescope participating in the night's observations. This list includes the observation time (in UTC time), the astronomical objects to be observed, their coordinates (typically in Right Ascension, R.A., and Declination, Dec.), and other relevant information. Observation control is typically managed by the Observation Control System (OCS[3]), which operates based on the provided observation list and additional parameters, such as the exposure time, number of exposures, required filters, telescope position and focus control. The images observed by the telescope are subsequently transferred to the data pipeline[4] for processing, extracting the photometric information required by astronomers, and identifying transients as high-value objects.

[1]University of Chinese Academy of Sciences, Beijing, China. [2]Key Laboratory of Optical Astronomy, National Astronomical Observatories, Chinese Academy of Sciences, Beijing, China. [3]School of Computing Science, Simon Fraser University, Burnaby, BC, Canada. [4]College of Science, Tibet University, Tibet, China. [5]Institute for Frontiers in Astronomy and Astrophysics, Beijing Normal University, Beijing, China. [6]Liii Network Co., Zhejiang, China.
✉e-mail: wangcunshi@nao.cas.cn; liyuyang22@mails.ucas.edu.cn

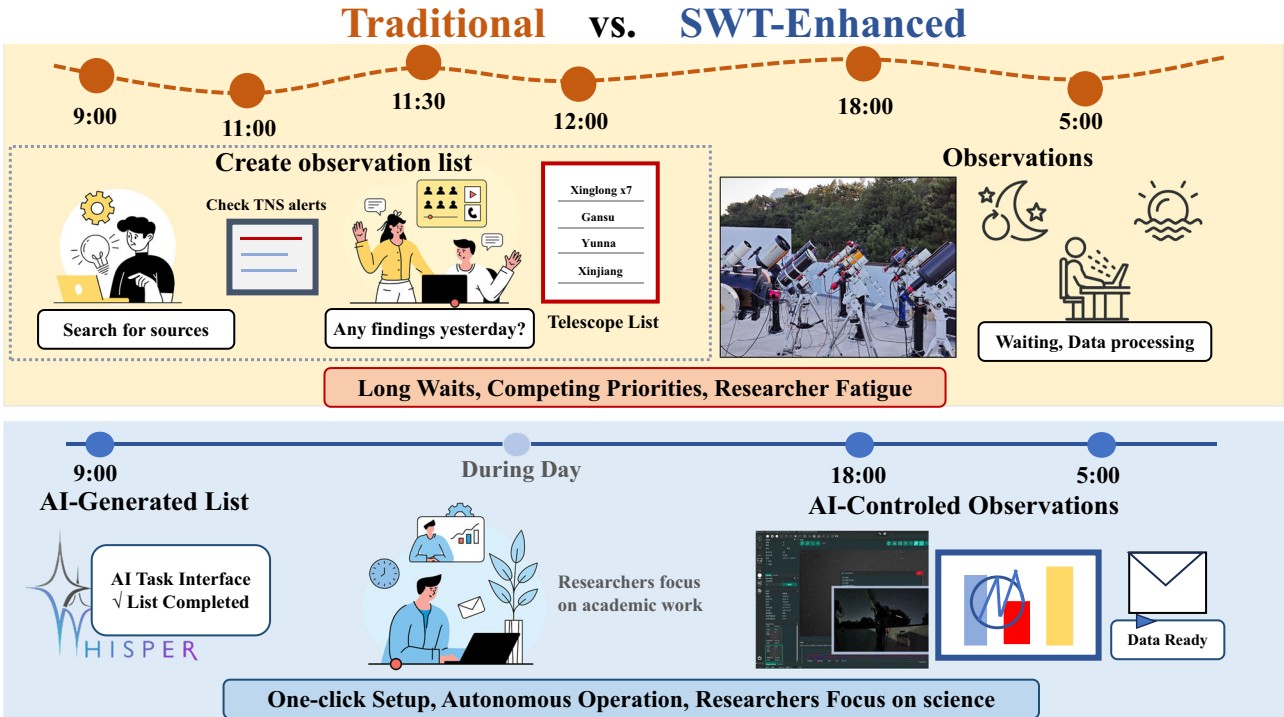

**Fig. 1 | Workflow comparison between traditional observation methods and the SWT enhanced system.** The TNS stands for Transient Name Server.

The observation, planning, and data processing phases are labor-intensive and require considerable effort. Generating the observation list involves selecting the most suitable objects from an input catalog containing more than 100,000 entries while taking into account the object's sky visibility at each telescope's position. Typically, each observation is first reviewed by the astronomer prior to being processed. The review process involves examining nearly 20GB of raw image data per telescope to detect and confirm the presence of transient astronomical events that are of interest to astronomers. To free astronomers from these phases with tasks, we developed the StarWhisper Telescope (SWT) system, a LLM[5,6] agent-based astronomical observation system designed to assist astronomers by efficiently managing observations across a telescope array and saving astronomers' time.

While LLMs have demonstrated extraordinary capabilities in many fields, traditional LLMs, such as Qwen[7,8], the GPT series[9], and DeepSeek[10], have not been equipped with tools or trained with specific data for astronomical observations, and more likely to present well-known celestial objects, no matter it can be observed or not (See Supplementary Section 1, Observation planning by Qwen). The SWT system addresses these challenges by integrating function calls to expand the capacities of LLMs and LLM-enhanced workflows to infuse domain expertise.

Function calls enable LLMs to interact with external tools or systems by recognizing predefined operations, extending their capabilities beyond text generation to perform tasks or retrieve real-time information. This functionality is developed through training on datasets pairing input instructions with expected outputs, enhancing the models' instruction-following abilities[8,9]. When equipped with function call capabilities, LLMs are often referred to as LLM-based agents, or simply agents in these paper[11-16].

LLM-enhanced workflow connects steps in a series or parallel configuration, incorporating tools, Application Programming Interfaces (APIs), and LLMs to handle complex, multi-stage tasks and do function calls. Compared to traditional workflows[17], it can process unstructured data, make flexible judgments, and adaptively link each step of the process. This approach increases the credibility and usability of the system through the operation and interaction logic of natural language, while maintaining robust input/output structures[11,15,18]. The workflow in this manuscript infers the LLM-enhanced workflow.

The SWT system is applied to the Nearby Galaxy Supernovae Survey (NGSS, Section "Nearby Galaxy Supernovae Survey") to help the observers save time (Fig. 1). The NGSS project is a time-domain survey with 10 amateur-level telescopes, with a simplified telescope organizational structure similar to GOTTA. In summary, the SWT observation module (Section "Methods") consists of observation planning (Section "Observation planning"), observation control (Section "Observation control") and a data pipeline (Section "Data pipeline"). Before and after observations, SWT provides recommendations to observers through automated messages (Section "Suggestions given by the AI agent"). We show the observation and execution of the SWT system in Section "Strengths and weakness", and discuss its strengths and weaknesses in Section "Strengths and weakness". The solutions to the weaknesses are discussed in Section "Future works". The related works are referred in Supplementary Section 2, Related Works.

## Nearby Galaxy Supernovae Survey

As outlined in Section "Introduction", we have developed an agent-based system named SWT, aimed at easing the challenges associated with astronomical observations, particularly focusing on large-scale time-domain surveys such as the GOTTA (More information are provided in Supplementary Section 3, GOTTA). Although GOTTA is still under development, we test the SWT system's performance through a similar survey called the NGSS, which features multiple sites and telescopes, thereby serving as an ideal testing ground for GOTTA.

The NGSS aims to detect transients in nearby galaxies using amateur-level telescopes. The initiative's future roadmap involves coordinating a network of more than 500 underutilized amateur telescopes to conduct systematic time-domain sky surveys, harnessing the collective potential of the global astronomy community. Similar to the GOTTA, its primary function is to identify transient astronomical events, but on a smaller scale, suitable for amateur equipment.

Several amateur telescopes have been installed at the Xinglong Observatory in Hebei Province (40.393° N, 117.575° E), with additional telescopes located in Xinjiang (43.522° N, 88.577° E), Gansu (35.678° N, 106.848° E), and Yunnan (23.914° N, 102.653° E) provinces in China. These telescopes, detailed in Table 1, are equipped with monochrome CMOS

**Table 1 | Details of amateur telescopes used for NGSS**

| Telescope name | Field-of-view (deg²) | Focal length (mm) | Pixel scale (arcsec pixel⁻¹) | Telescope location |
|---|---|---|---|---|
| xl-106 | 10.14 | 530 | 1.46 | Xinglong |
| xl-130 | 2.91 | 990 | 0.784 | Xinglong |
| xl-130-2 | 3.43 | 909 | 0.853 | Xinglong |
| xl-180 | 4.80 | 502 | 1.54 | Xinglong |
| xl-203 | 1.18 | 864 | 1.29 | Xinglong |
| xl-250 | 0.18 | 2575 | 0.301 | Xinglong |
| xl-c14 | 0.16 | 2717 | 1.14 | Xinglong |
| gs-150 | 0.83 | 678 | 0.73 | Gansu |
| yn-90 | 2.27 | 601 | 1.85 | Yunnan |
| wlmq-107 | 2.47 | 700 | 2.8 | Xinjiang |

The telescope name contains two parts, 'xl', 'yn', 'gs', 'wlmq' is the place, and the rest shows the diameter or the model number of the telescope.

sensors and LRGB photometric filters. With their relatively large Field of View (FoV) and apertures under 250 mm, they are particularly suited for detecting bright supernovae in nearby galaxies. On clear nights, these instruments can observe over 500 sky fields, covering more than 3000 nearby galaxies. Larger amateur telescopes with apertures exceeding 250 mm and narrower FoVs are used primarily for follow-up photometry, complemented by professional telescopes like the 216-cm telescope at Xinglong Observatory for follow-up spectroscopy and precise photometric observations. Considering the power of these telescopes, this work focuses on galaxies within 50 Mpc[19] and associated transient phenomena.

To reflect the operational preferences of the amateur astronomy community as much as possible, NGSS operations utilize Nighttime Imaging 'N' Astronomy (N.I.N.A.) software, a widely-used automation tool among amateur astronomers. This software integrates the control processes, allowing the observer to connect all equipment, such as the camera, telescope, dome, and filter wheel, to execute the observation from a given list in a specific format (a ninaTargetSet file). N.I.N.A. allows users to develop custom plugins to meet specific needs, for example, the three point polar alignment. The software requires a list file to execute observations and uses ASCOM drivers to control the entire telescope system, performing fully automatic operations including slewing, auto-focusing, astrometric solving, capturing images, switching targets, and returning to park position upon completion.

## Methods

LLM-based agents relieve researchers of repetitive tasks in scientific workflows, enabling a strategic shift toward high-impact intellectual contributions. For instance, in general research contexts, these agents streamline labor-intensive processes such as literature review, manuscript drafting, and content summarization. In astronomy, specialized agents further augment productivity by automating critical but there are time-consuming responsibilities, including preparing observation proposals, designing and executing observational strategies.

The SWT system is designed to orchestrate and interface with the NGSS telescope network through AI-driven agents, enabling autonomous control and adaptive observation workflows. Several functional workflows are interconnected with workflows and API-driven communication protocols.

In this section, we present the architecture of the SWT system, which comprises a central agent (Section "Central agent") and four sequentially connected workflows (Fig. 2): observation planning (Section "Observation planning"), telescope control (Section "Observation control"), data processing (Section "Data pipeline") and agent reporting (Section "Suggestions given by the AI agent"). The central agent is integrated with all workflows, providing comprehensive control over each component.

Regarding the LLMs used in the SWT system, agents and workflows, we utilize Qwen-2.5[20] as our base model. The details of the architectures are

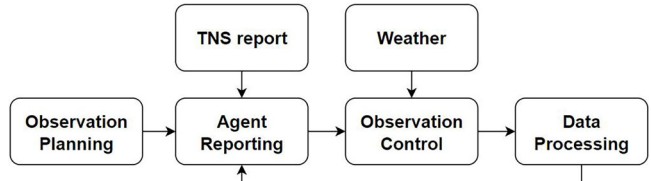

**Fig. 2 | The main structure of the SWT system.** It begins with Observation Planning, where initial observation lists are generated. The process then moves to Agent Reporting, which incorporates updates from the Transient Name Server (TNS) report and the transients found by the data processing module. This procedure also allows manual revision and injects new targets into the observation list. These plans are then operated by the telescopes via the Observation Control module, in which the weather station can send commands to halt the observation. The Data Processing phase is designed to find transients from a data pipeline.

provided in the Supplementary Section 4, Detail and Prompt Design of NGSS Agent; Supplementary Section 5, Observation Planning and Supplementary Section 6, Auxiliary tools.

### Central agent

The SWT system adopts a modular workflow architecture designed to automate astronomical observation tasks while maintaining flexibility for human intervention. Built upon a distributed processing paradigm, the SWT system integrates 8 specialized workflows that operate through coordinated API communications, parameter-driven task routing, and AI-enhanced decision layers.

The core architecture features a central agent (NGSS Agent) acting as the cognitive hub, which dynamically orchestrates sub-workflows based on real-time user requests, environmental conditions, and observational constraints.

The eight workflows and their structure are as follows:

- Central Workflow: The central workflow orchestrates the overall system operations, including setting parameters, managing memory buffers, handling LLM models, and invoking tools via a central Agent.
- Observation Planning: This workflow generates observation plans, communicates with the local server, and produces a log Uniform Resources Locator (URL) and the task Universally Unique Identifier.
- Observation List Query: This workflow allows users to query observation plans for a specific site and date.
- Transient Loading: This workflow handles the viewing and addition of transient sources and provides links to identification and image results.
- Target Addition: This workflow enables users to add new astronomical targets (including targets from Transient Name Server, TNS) to the observation list and provides a summary of the results.
- Plan Loading: This workflow loads observation plan files into the telescope's N.I.N.A. software.
- Telescope Control: This workflow controls the start and end of telescope observation actions via API calls.
- Weather Monitoring: This workflow retrieves and summarizes weather data to ensure suitable observing conditions.

We show the details and the prompt design for the main agent in Supplementary Section 4, Detail and Prompt Design of NGSS Agent.

### Observation planning

The observation planning workflow is designed to create observation lists for 10 telescopes across four sites in NGSS. Further details of this workflow can be found in Fig. 3.

The observation planning is done by a function call to trigger the workflow, and sequentially trigger the generation of a day's plan. The workflow first generates an empty observation list based on the configuration files. The configuration file includes the pre-set observation parameters, including but not least the exposure, tracking, focusing and guiding settings,

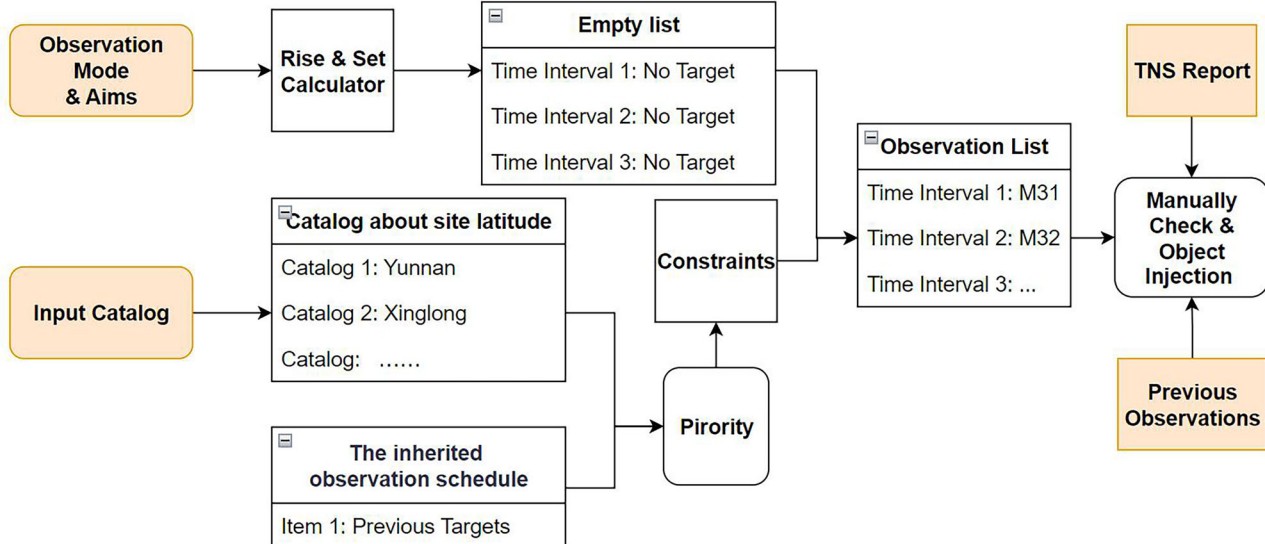

**Fig. 3 | Schematic diagram illustrating the construction of the observation list for the NGSS.** Regular rectangles denote tools, while rounded rectangles represent agents powered by LLM. All orange boxes indicate input information. The constraints tool calculates both altitude and moon distance limitations. Once objects are manually added, the finalized list is saved to the server.

filter configurations, slewing and readout time, is designed to improve the workflow's robustness. The workflow then fills the empty observation lists with sources in the input catalog, or the previous observation lists. The site position and several constraints will be considered during planning, and the final plan will be revised by the astronomer.

More details of Observation Planning are shown in Supplementary Section 5, Observation Planning.

## Observation control

After the generated observation list is reviewed by an astronomer, it will be transferred to the appropriate format and sent to N.I.N.A. to start the observation (Fig. 4).

We built a N.I.N.A. plugin based on the Site Plugin[21] (https://nighttime-imaging.eu/docs/master/site/contributing/plugins/) with the functions of loading ninaTargetSet, an observation list format that can be read by N.I.N.A., and starting and stopping observation via a User Datagram Protocol message. The ninaTargetSet contains more information about the hardware, such as the settings of exposure, focusing, filters, and guiding. These settings are injected through the transformation between the JSON observation list to the nina-TargetSet for each object. By default, every two hours, the automatic focusing will be enabled, and 30 new bias images will be taken.

At the predetermined time (the end of astronomical twilight), the agent issue commands by sending corresponding messages, which then triggers the plugin to initiate observations. Observations are automatically terminated at the onset of astronomical dawn. Additionally, the API from the compact weather site is integrated with the agent, enabling the automatic suspension of observations in response to adverse weather conditions.

## Data pipeline

The raw image taken by the telescopes will be processed in the data pipeline to find transients and send alerts. Usually, the follow-up spectroscopy observation proposal will be sent to the 216-cm telescope (at Xinglong Observatory), and manually do the follow-up observation since its not linked to the SWT system.

Xinglong - Observatory Popular Science Telescope Pipeline (X - OPSTEP,[22]) is the data pipeline for NGSS. It can automatically and real-time process the raw image data, including image pre-processing, astrometry, photometry, and image subtraction.

After the observation process by N.I.N.A., the raw image will be sent back to the server at the Xinglong observatory, then X-OPSTEP will be triggered as long as the new raw data is received. X-OPSTEP first performs bias and flat correction, and then solves for the image world coordinate system information with `Astrometry.net`[23]. After that, the photometry process will be done by SExtractor[24], and the resulting catalog will be cross-matched with GAIA DR3[25] to conduct flux calibration. The pipeline will then combine the same target for the same band By using SWarp[26] with 3-$\sigma$ clipping median combination to get rid of contamination from any satellites. Finally, we use `HOTPANTS`[27] to subtract the template image from the science image and cut out the target galaxies. See Fig. 5 for an example. A real-bogus model is applied to check the subtracted image. The model, similar to Shi et al.[28], using a $64 \times 64$ difference image as input, predicting through a ResNet + attention architecture, achieves 99.12% validation accuracy.

## Suggestions given by the AI agent

The photometric data and image subtraction result will be read by the SWT system. The SWT system evaluates multiple options to assign, remove, or replace specific galaxies in the observation lists. Typically, it recommends that the user adopt the observation lists from the previous day to form continuous monitoring of these galaxies. These galaxies are flagged and reassigned to the observation list. Similarly, galaxies with detected transients are also highlighted and included in the subsequent observation list for continuous imaging.

Once a transient is detected, its coordinates and associated information are transmitted to the user via a web interface. If the Agent is unable to retrieve any information about the object, it will recommend that the user submit the object to the TNS. The SWT system will also send a message about the transient to the XingLong chat group, including information about the source and an observation suggestion generated by the LLM.

We show some auxiliary tools in Supplementary Section 6.

## Results

The SWT system has been actively operating on NGSS telescopes since October 2024, successfully detecting several transients. The majority of these transients are initially reported by other open surveys, and we cross-matched them with our observational data, and found some transients bright enough to be detected. Some transients have observing time earlier than the discovery time reported, but not detected due to large background noises or detected before explosion. Details of these transients are shown in Table 2.

As shown in Table 2, during the early test observations, the actual detection time for transients lagged approximately one day behind the TNS

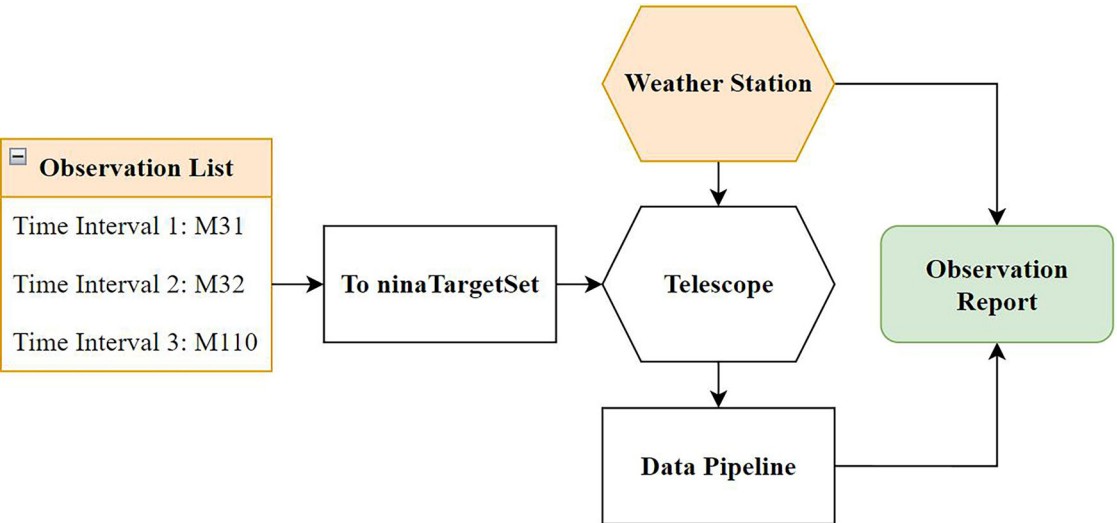

**Fig. 4 | Schematic diagram of the observation component in the NGSS.** Regular rectangles denote tools, while hexagons represent hardware facilities. All orange boxes indicate input information, and green boxes correspond to output results. The observation list will be transformed into a ninaTargetSet file for N.I.N.A. to read and take control of the observation. The data will be simultaneously processed by a data pipeline. The weather station at Xinglong Observatory can automatically halt the observation by commands to N.I.N.A. The transients found will be reported after data processing.

OBJNAME:NGC0615
DISTANCE:24.234375 Mpc
BMAG:12.15625
x=5768,y=6032,width=221 pixel
ra=23.7736455,dec=-7.3402917
hms: 1:35:5.675,dms: -7: 20: 25.05

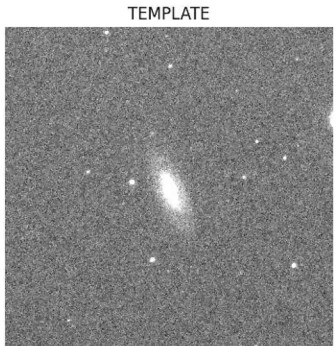
TEMPLATE

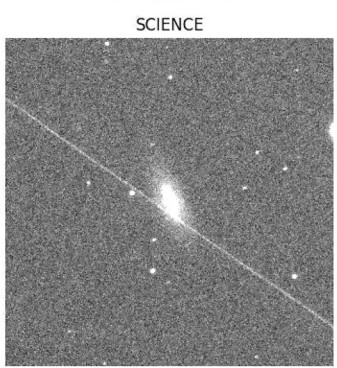
SCIENCE

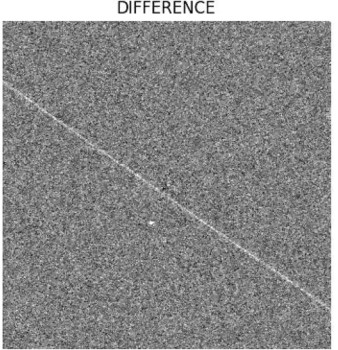
DIFFERENCE

**Fig. 5 | The discovery confirmation images of AT2025pk.** The images are shown from left to right are template, science, and difference image. The title contains object name (OBJNAME), background magnitude (BMAG), the pixel position of AT2025pk, Right Ascension (ra), Declination (dec), and the hour angles in hour-minute-second (hms) and degree-arcminute-arcsecond (dms).

**Table 2 | Details of transients detected by NGSS**

| Transient name | Our first detection | TNS report date | Transient type | Redshift (z) | Discovery telescope |
|---|---|---|---|---|---|
| SN2024xin | 2024-10-07 | 2024-10-05 | SN Ia | 0.019 | xl-106 |
| SN2024xlh | 2024-10-07 | 2024-10-06 | SN II | 0.03 | xl-106 |
| SN2024xli | 2024-10-07 | 2024-10-06 | SN Ia | 0.037 | xl-106 |
| SN2024xqe | 2024-10-10 | 2024-10-09 | SN Ia | 0.034 | xl-106 |
| SN2024xvg | 2024-10-10 | 2024-10-10 | SN Ia | 0.03 | xl-250 |
| AT2024abqt | 2024-11-21 | 2024-11-21 | CV | – | xl-130-2 |
| SN2024advj | 2024-12-11 | 2024-12-11 | SN IIn | 0.017 | xl-106 |
| SN2025bl | 2025-01-04 | 2025-01-04 | SN II | 0.009 | xl-130-2 |
| AT2025pk | 2025-01-17 | 2025-01-17 | Flare candidate | – | xl-130-2 |

The first detection date is the date on which the transient was detected based on the image subtraction. The '–' in redshift means no data reported. The TNS is the Transient Name Server plateform, and the types are Super Novae (SN), Cataclysmic Variable (CV).

**Table 3 | Comparison between manual and SWT system for observation planning**

| Metric | Manual (PhD student) | SWT system |
|---|---|---|
| Planning time per telescope | 1–1.5 h | <1 min |
| Number of galaxy coverage on the same day | 2000–2500 | 2500–3000 |
| Conflict rate per list | 1–3 | 0 |
| Requires manual adjustment | Frequently needed | Not required |
| Reproducibility/Consistency | Depends on operator | Fully deterministic |
| Adaptability to constraints | Limited | Sites, visibility, and moon phase |

The manual plan is generated by a PhD student, with galaxy coverage based on the combined 10-telescope array. The conflict rate reflects the number of galaxies observed more than once in a single list. The time refers to the period when observable galaxies can be distributed across the telescopes. Manual adjustment of the SWT plan is only applied when target changes are needed. In manual planning, it is difficult for humans to account for all constraints simultaneously.

reporting time. This delay is primarily attributed to the manual maintenance at the beginning of NGSS, at the time when the manpower can only afford two to three telescopes.

However, after deploying SWT system, the detection time of brighter transients, such as AT2024abqt, SN2024advj, and SN2025bl, trailed the TNS reporting time by only a few hours. Notably, AT2025pk, a flare star, represents the first transient event successfully identified through the SWT system.

Figure 5 shows the transient source identification images provided by the SWT system. Bright transient sources are evident in both the science and difference images. Through visual inspection, we search for suspicious bright spots in the images and cross-check to exclude asteroids or comets, thereby enabling the discovery of transients.

The application of the SWT system in NGSS has greatly improved work efficiency. The observation plan was completely generated manually in the beginning phase of NGSS. To quantitatively evaluate the performance of the SWT system, we conduct a controlled comparison between manually generated and SWT-generated observation plans under identical conditions. The observation site is considered as Xinglong. The results (Table 3) show that SWT reduce the planning time from about 1.5 h (PhD level) to less than 1 min, with better target coverage counts and zero conflicts, demonstrating both efficiency and robustness.

We evaluated the success rate of SWT system by cycling through the following prompts and querying the agent round by round: (1) Please make an observation plan; (2) Please check today's log of making observation plan; (3) Please check the observation plan; (4) Please check the transients; (5) Please add them to the observation list; (6) Please load the observation lists to each telescope. These queries are commonly used among observers when interacting with the agent. The programs for starting or stopping observations have been prohibited from running during the test to avoid any potential risk of system damage.

The agent has been asked 7620 rounds, of which 4194 involved tool calling, and the tool has been called 2962 times successfully, showing the overall function call success rate is ~70.5%. The number of function calls is less than the number of queries because not all queries need to make function calls. The success rates of the observation planning and observation list querying tools have achieved 100%. For loading transient or input sources and telescope-controlling tools, the success rates range between 60% and 70%. The loading observation plan for the telescope tool has a relatively low success rate of ~30%. These issues are mainly caused by the network latency, and lead to an overtime failure. The test is conducted over two days, and the total running cost is about 871.6 min, involving about 58.6 million tokens cost. Tokens are a basic unit of text processed by LLMs, providing the context for usage in our workflow. This amount of tokens is about 14 dollars at cloud services.

The example of usage of the SWT is shown in Supplementary Section 7.

## Discussion

In the previous Section "Methods", we show the construction of the SWT system. In this section, we first introduce the strengths and challenges of the system in Section "Strengths and weakness". We then make a future plan at Section "Future works" to solve the technical issues we faced, and the path to an AI Astronomer.

### Strengths and weakness

Astronomical observations have traditionally demanded specialized expertise; however, our agent-integrated system substantially lowers the entry barrier by enabling natural language interaction and real-time process monitoring. This innovative approach not only accelerates the onboarding process for novice observers but also empowers experienced astronomers to efficiently manage multiple telescopes simultaneously, thereby considerably improving observational productivity.

The development of standardized protocols, such as Agent-to-Agent[29], which allow agents to interact with another agent, among different agents, lays a solid foundation for cluster control designed based on multi-agent system games. It enables seamless customization for a wide range of survey projects and adaptability to various observational strategies, including dynamic updates to observation lists and real-time control.

We also identify several limitations of the current system and discuss how they may be addressed or what lessons have been learned.

**Automated hardware.** The dome of NGSS is not automated and must be opened or closed manually, rendering automatic observation halting partly ineffective. At Xinglong Observatory, this issue is mitigated by sending warnings to the observers of the GOTTA pathfinder, which shares the same dome. However, for telescopes located at other sites - typically housed in amateur domes situated on villagers' rooftops - dome operations cannot be controlled remotely. Dome management at these locations relies entirely on the cooperation of local residents who rent out their rooftop space. Similarly, flat exposure must currently be taken manually, due to the lack of automation. At present, we have created standard template flat images to ensure basic data quality in Gansu, Yunnan, and Xinjiang telescopes. Looking forward, integrating automated hardware will be a feasible solution for telescopes at Xinglong Observatory. However, for remote amateur sites, achieving full automation remains a considerable challenge due to logistical and infrastructural constraints.

**Hardware failures.** The SWT system is currently unable to automatically resolve hardware failures. Common issues include focuser freeze due to cold weather, wiring disconnections, and computer memory crashes, with the latter two being the most frequent. At present, these problems are resolved manually by on-site observers. In the future, the system could be enhanced to provide troubleshooting suggestions through an intelligent agent. This would be achieved by analyzing observation logs written by observers, which the SWT system could access using Retrieval-Augmented Generation[30] or a Model Context Protocol. Retrieval-Augmented Generation is a technique that enhances LLM by retrieving information from a specified dataset, considerably improving accuracy and relevance in specific domains. Model Context Protocol is a protocol designed to define contextual interactions for tools or datasets, aiming to standardize communication and integration between models and external resources. For larger telescopes, continuous monitoring systems could be implemented to collect standardized telemetry data, allowing the agent to assess telescope status and detect anomalies in real time. However, for telescopes located outside of Xinglong Observatory - particularly those hosted in amateur setups - there are currently no feasible conditions for immediate or automated fixes due to limited infrastructure and remote locations.

**Software crashes.** The SWT system currently lacks the ability to automatically resolve low-level software crashes, such as those occurring

in applications like N.I.N.A. and X-OPSTEP. These incidents are typically infrequent but often stem from human-driven overuse of server resources-such as running multiple memory-intensive programs simultaneously, which exceeds available storage capacity. Automated solutions for these types of failures are not considered a high priority, mainly due to their rarity and the challenges associated with changing operator workflows or upgrading server specifications. While agent-based monitoring of the software environment could potentially detect and mitigate such issues, it introduces considerable security risks by requiring privileged access (e.g., through tools like Code Interpreter). An alternative approach involves training a GUI agent[31] to monitor graphical interfaces and respond to anomalies. However, this method requires substantial GPU memory resources, making it impractical for deployment at scale in large observatory operations.

**Standardized telescopes.** The current lack of standardization among telescopes in the NGSS network introduces operational inconsistencies, particularly in manual observation practices. Two key challenges emerge for the SWT. First, the varying FoVs across different telescopes affect overall sky coverage. The input catalog is currently designed based on the smallest FoV to maintain uniformity, which results in larger FoV telescopes potentially observing adjacent or overlapping regions and inadvertently covering galaxies already targeted by smaller FoV instruments. This leads to a reduction in total effective sky coverage compared to ideal manually generated plans. A new tool is being developed to shift the planning focus from individual galaxy selection to maximizing sky coverage, which is expected to improve this limitation. Second, differences in hardware reliability across telescope types result in varied failure patterns, making it more difficult to implement consistent fault detection and recovery strategies. Standardization of both hardware and software components is therefore essential not only for improving system usability and agent-instrument interaction, but also for enabling scalable deployment. As the NGSS network expands with more standardized nodes, it will achieve broader sky coverage, enhancing its scientific and educational capabilities. Furthermore, standardization reduces operational complexity by minimizing labor-intensive maintenance and improves data consistency by reducing instrumental variation across observational outputs.

## Future works
**Edge computing.** Integrating edge computing modules with telescope systems offers a promising solution to latency issues in server-to-server communication, enabling telescopes to make autonomous decisions and improving operational responsiveness and control stability. By deploying lightweight agent services and AI models locally-such as on industrial personal computers equipped with edge devices like the NVIDIA Jetson-real-time decision-making and task execution can be achieved with minimal reliance on remote servers or cloud infrastructure.

**Observation monitoring.** A future observation monitoring module is crucial for ensuring the reliability of unattended observatories. It will use sensors to collect real-time data on the telescope's physical state, including vibration (e.g., sounds from the equatorial mount), temperature, humidity, and electronic signals, to assess its operational status. When an anomaly is detected, the module will evaluate its severity and respond accordingly-automatically resolving low-severity issues through control commands or warning the observers with guidance for more serious problems.

We will collect operational data from the GOTTA prototype, a 1-m diameter prototype telescope for GOTTA (See more information in Supplementary Section 3, GOTTA), to generate training samples for various types of errors and malfunctions. These data will be used to construct a knowledge graph based on observer handbooks, aiming to enable automated monitoring and maintenance instructions-capabilities that are currently limited in existing systems. This effort will also include building a comprehensive dataset containing videos from cameras near the telescope, observation logs, and hardware manuals, which will serve as valuable training material for LLM-based agents assisting in system maintenance.

This dataset serves as a Retrieval-Augmented Generation-based memory module, integrating operational logs, interaction histories with observation assistants, and user behavioral profiles. These multi-source data are transformed into vector embeddings, enabling the retrieval of contextually relevant information during inference. By continuously learning from past successes, failures, and user-specific patterns, the system dynamically refines its responses, adapts to individual preferences, and supports distributed intelligence across edge computing nodes (e.g., telescope-mounted devices).

**Path to AI Astronomer.** Figure 6 shows the methodology used by the AI Astronomer. For the telescopes that conduct sky surveys, it will get data from their real-time observation, and command the follow-up spectroscopy telescope to observe.

In Fig. 6, the red blocks represent AstroInsight, an idea generation system which built on the integration of LLM-based agents and astronomical datasets. It will generate scientific ideas based on knowledge from databases such as ArXiv or Astrophysics Data System, and selects optimal methodologies for validation.

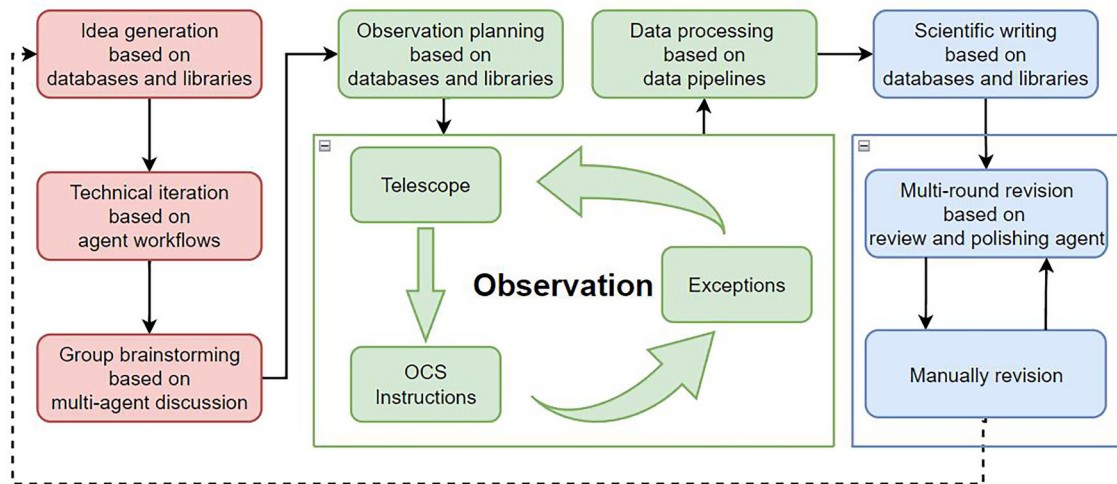

**Fig. 6 | The iterated process of AI Astronomer.** The red blocks are about the StarWhisper automatic scientific discovery system. The green blocks are the observation part, where we are trying to build the embodied telescope. The blue blocks show the StarWhisper scientific writing workflow. The OCS stands for the Observation Control System.

For feasible plans requiring new data, the green blocks-enhanced by reasoning models like DeepSeek R1[32]-support operations across the NGSS project, GOTTA, and 216-cm-telescope. This integration enables reasoned task dispatch and observation coordination, with acquired data feeding directly into AutoASTRO, the automated data processing and machine learning pipeline. The blue blocks represent the StarWhisper's scientific writing workflow, which transforms analytical outputs into structured narratives. This workflow forms a closed-loop feedback system, where the results of scientific writing are fed back into the idea generation module (red blocks), enabling continuous iteration and knowledge refinement.

## Data availability

We have uploaded the codes, prompts, tools, and supporting materials to GitHub at https://github.com/Yu-Yang-Li/StarWhisper/tree/main/NGSS. The StarWhisper's scientific writing workflow can be found in the Coze store at https://www.coze.cn/store/agent/7383556199616135179?utm_source=ai-bot.cn. See https://www.coze.cn/store/agent/7383556199616135179?utm_source=ai-bot.cn&bid=6f7h1ou4k7g19&post_id=7396577135529394202 for the user handbook.

## Code availability

See https://github.com/Yu-Yang-Li/StarWhisper/tree/main/NGSS for the usage of the codes.

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

## Acknowledgements

The research presented in this paper was generously funded by the National Programs on Key Research and Development Project, with specific contributions from grant numbers 2019YFA0405504 and 2019YFA0405000. This research is supported by National Key R&D Program of China (grant No. 2023YFA1609700 and 2023YFA1608304). Additional support came from the National Natural Science Foundation of China (NSFC) under grants NSFC-11988101, 11973054, 12090040, 12090041, 12403022 11903054, 12261141690 and 11933004. We also received backing from the Strategic Priority Program of the Chinese Academy of Sciences, granted under XDB41000000, XDB0550000, XDB0550100 and XDB0550102. Special acknowledgment goes to the China Manned Space Project for their science research grant, denoted by NO.CMS-CSST-2021-B07. The research is also supported by National Astronomical Observatories Chinese Academy of Sciences No. E4TG2001 and Key R&D Program of Zhejiang (2024SSYS0006). J.F.L. extends gratitude for the support received from the New Cornerstone Science Foundation, particularly via the NewCornerstone Investigator Program, and the honor of the XPLORER PRIZE.

## Author contributions

Cunshi Wang contributed the coding, agent construction, and the writing of the manuscript. Yu Zhang and Xinjie Hu contributed the coding, agent

construction. Yiming Mao contributed the test on NGSS project. Xunhao Chen contributed the X-OPSTEP pipeline. Pengliang Du contributed the data transmission and connections between telescopes to agents. Rui Wang contributed setup and operation of the telescopes. Yuyang Li contributed the design of the structure, prompt engineering for the SWT system. Ying Wu contributed the design NGSS array and the design of N.I.N.A. plugin. Hang Yang contributed the edge calculation of the system. Yansong Li contributed the writing of the manuscript. Beichuan Wang contributed the TNS search module and the observation of these telescopes. Haiyang Mu, Xiaohan Chen, Shunxuan He, Hao Mo, Liyue Zhang, Lin Du, and Yunning Zhao contributed the operation and maintenance of SWT system on NGSS. Jianfeng Tian, Liang Ge, Shengming Li, Xiaomeng Lu contributed the design of the structure on embodied telescope and the future of SWT system. Zheng Wang, Jinhang Zou contributed the design of GOTTA pathfinder OCS and provide insights in the observation based on agents. Yongna Mao, Yang Huang, Ningchen Sun, Jie Zheng, Min He, Yu Bai, Junjie Jin, Hong Wu, Jifeng Liu contributed the support the SWT system and administrative and resource support.

## Competing interests

The authors declare no competing interests.
