## [Transparent Peer Review file · Communications Engineering]

StarWhisper Telescope: Agent-Based Observation Assistant System to Approach AI Astrophysicist

Corresponding Author: Dr Cunshi Wang

Version 0:

Reviewer comments:

Reviewer #1

(Remarks to the Author)

This work presents StarWhisper, an agent-based observation assistant for astronomy. It puts into practice many recent ideas regarding agentic AI applied to discovery and follow-up of nearby galaxy supernovae based on a network of small aperture telescopes. The main claim of this work is to establish a new paradigm of embodied AI systems to accelerate discoveries in astrophysics.

The authors are experiencing with new AI tools that will change how astronomy (and most natural sciences) are currently done, and the exact combination of tools that are being used is useful information for the community. However, almost no effort is done to show data and evidence about how this system works. Also, very little is done comparing current efforts that achieve much faster reaction times than those discussed in this work (a few minutes) without using AI agents.

To merit publication I believe that apart from reporting the workflow in detail, which is very useful, a lot more evidence about the performance of the system should be provided, e.g., execution time statistics, failure rates, fraction of cases that required human intervention, and, crucially, relevant metrics that show an improvement over non AI agent-based system (if one claims to accelerate discoveries, evidence should be provided that this is the case).

I tend to agree with the authors that the future of astronomy will look like what they have done, and I congratulate them for their work. However, the evidence is not there.

Reviewer #2

(Remarks to the Author)

The paper presents work that aims to automate the process of detecting transients, preparing target lists and scheduling telescopes using AI agents that then execute the wishes of the astronomer. The work is novel in its use of large language models as the interface between humans and AI agents. It will be of interest to astronomers, software engineers, and those with an interest in artificial intelligence (AI).

The authors report on the work they have done in developing and testing these AI agents on an array of amateur telescopes that are currently part of a project to detect transient astronomical phenomena (e.g. supernovae) in nearby galaxies. The paper presents results from approximately 5 months of testing. The authors demonstrate that AI agents (with some human interaction) can successfully discover astronomical transients once instructed by astronomers.

The authors then describe their plans for the future. In particular, extending the role that the AI agents play in other aspects of the discovery of transients and astronomy with the aim of improving the productivity of astronomers.

The paper is not acceptable in its current form.

Some of the terminology is unclear. Examples include: "a limiting observational personnel sequence" in the introduction, "industrial computer memory explosion," in section 5 and "directional position deviation" in section 6. There are also awkward phrases, grammatical errors, incomplete sentences, typos (e.g., "amputator-level" which should presumably be "amateur-level"), and spelling mistakes. I recommend that the journal use the services of a language editor to help the authors express their ideas more clearly.

There is some jargon, for example, "AI agent workflow" and some less well-known acronyms are not defined, e.g. UUID. Not everyone will be familiar with UUID and in this particular case, the definition used in Listing 1 (on p. 30) differs from what I

had understood this acronym to mean. If one were to Google this acronym, then one would find Universally Unique Identifier.

The paper can be reduced significantly by removing some of the repetitive text and focussing on what has been achieved rather than what is planned.

Reviewer #3

(Remarks to the Author)

1. Summary of the Article's Main Findings

This paper introduces the StarWhisper Telescope (SWT) system, an agent-based AI framework designed to automate and optimize astronomical observations for time-domain surveys such as the Nearby Galaxy Supernovae Survey (NGSS). The authors claim that, by integrating large language models (LLMs) with function-calling capabilities, real-time pipelines, and modular agent workflows, the SWT system can autonomously generate site-specific observation lists, execute real-time image analysis, and dynamically trigger follow-up actions. They report a reduction in human intervention of up to 90%, specifically for target prioritization, scheduling, and data reporting. Deployed across NGSS's network of amateur telescopes, SWT has achieved a median discovery lag of less than 12 hours for nearby transients. The system is presented as a prototype for future AI-driven telescope networks, such as the planned SiTian Project.

2. Significance

High.

The manuscript addresses a critical bottleneck in time-domain astronomy: the labor-intensive nature of scheduling, executing, and reducing data from large telescope networks. Automating these functions with LLM-based agents is both timely and visionary. The significance of this work lies in its demonstration that agent-based systems powered by LLMs can meaningfully reduce human intervention while maintaining scientific utility—particularly in the context of modest, amateur-grade instruments.

The authors report a median discovery lag of less than 12 hours for nearby transients, which is a meaningful benchmark. This outperforms typical amateur workflows, which often rely on manual coordination and may detect transients with lags of 1–3 days. Moreover, this performance is competitive with—or better than—some professional surveys, such as the Zwicky Transient Facility (ZTF) and OGLE-IV, which generally have revisit cadences of 1–4 days.

This work could influence how next-generation observatories and surveys—such as SiTian or even LSST—approach AI integration.

3. Novelty

Moderate to High.

While autonomous observation systems and robotic observatories are not new, integrating a function-calling LLM-based agent for end-to-end control—including observation planning, execution, real-time data reduction, and feedback—is a novel and ambitious concept. The scale at which the authors aim to deploy this framework (hundreds of telescopes, hybrid edge/cloud setups) sets it apart.

4. Technical Soundness

Generally sound, but with caveats.

- The architecture and modularity are well described, and the individual components (e.g., observation planning workflows, control plugins, data pipeline) are logically structured and practically implemented—at least within the NGSS use case. The use of open-source tools (e.g., N.I.N.A., ASCOM, Astrometry.net, SExtractor, SWarp, HOTPANTS) reflects sound engineering practices and supports reproducibility.

However, quantitative performance evaluations are relatively weak for a paper proposing a system of this scope. Key limitations include:

- Discovery lag is discussed in terms of anecdotal comparisons rather than systematic benchmarking. In fact, the claim of a median discovery lag of less than 12 hours for nearby transients appears only in the abstract, with no supporting data or detailed analysis in the main text.
- The paper lacks a systematic evaluation of the system's impact: for example, the only quantitative comparison provided is that the assistant previously spent 2–3 hours per night preparing observations manually, time which is now saved through automation. However, this is presented without supporting metrics or an ablation study comparing performance with and without the agent-based architecture.
- Error rates from real/bogus classification are not reported.
- Known failure modes (e.g., hardware crashes, cold weather issues, software faults) are acknowledged, but no quantitative analysis or mitigation plan is presented.
- Some references (e.g., “[2], [3] in prep.”) are placeholders and do not allow independent verification.

In summary, while the system appears to work as intended in its current limited deployment, the paper would benefit from stronger empirical evidence to support its claims of efficiency, scalability, and robustness—especially given its ambitions for widespread deployment. Notably, Section 6.1 (Strengths and Challenges) is relatively brief compared to the detailed technical descriptions elsewhere in the manuscript (e.g., system design and future plans), and it does not sufficiently expand on limitations, trade-offs, or lessons learned from real-world operation.

5. Clarity of Presentation

Moderate.

The manuscript presents a substantial amount of technical detail, and the structure is mostly coherent, but the clarity of writing and flow could be significantly improved. Specific issues include:

- The English is often awkward, with grammatical errors and inconsistent phrasing throughout. Many sentences would

benefit from language editing to improve readability and professionalism (see detailed list below).

- Some sections are overly dense, mixing implementation-level detail with conceptual discussion (e.g., system architecture and workflow descriptions), making it hard to follow the main thread.
 - Figures are generally helpful but would benefit from more descriptive captions and consistent visual formatting.
 - The length of the manuscript (including appendices) may exceed what is typical for Nature-style articles and could be better organized by moving technical implementation details to supplementary material.
- Despite these issues, the authors successfully convey the main idea, and with revision, the paper could be significantly more accessible and impactful.

6. Strengths

- Practical deployment: SWT is already operational in the NGSS network.
- Integration of open-source tools: N.I.N.A., ASCOM, Qwen, and X-OPSTEP are well integrated.
- Forward-looking vision: The ambition toward embodied telescopes and AI astrophysicists is thought-provoking.

7. Weaknesses

- Lack of rigorous evaluation: Key performance claims, such as the 12-hour median discovery lag and 2–3-hour daily time savings, are presented without quantitative validation or systematic benchmarking.
- Limited discussion of real-world limitations: While the authors acknowledge hardware and environmental issues (e.g., cold weather, N.I.N.A. crashes), these are not explored in depth, nor are mitigation strategies proposed.
- Imbalance in content: Significant effort is devoted to describing the architecture and future plans, but the section discussing operational challenges and lessons learned (Section 6.1) is disproportionately short.
- Writing quality: The manuscript contains frequent grammatical and phrasing issues, which hinder clarity. Some sections are difficult to follow due to dense technical descriptions and inconsistent structure.

8. Recommendation

Major Revision

The manuscript presents an ambitious and potentially transformative system for autonomous telescope operations using LLM-based agents. The integration of AI-driven planning, control, and real-time data analysis into a unified framework is timely and forward-looking, and the practical deployment within NGSS adds credibility.

However, the paper currently lacks the empirical depth, clarity, and balance expected for a journal of this level. The authors should provide stronger quantitative evidence to support their performance claims, expand the discussion of operational challenges, and revise the manuscript for clarity and language quality. If these issues are adequately addressed, the paper would represent a meaningful contribution to the field.

9. Detailed Comments

See attached PDF.

Version 1:

Reviewer comments:

Reviewer #2

(Remarks to the Author)

The authors have submitted a more concise version of the paper describing their work. Much of the material has been moved to appendices, which makes the paper more readable.

While there are still some grammatical errors and awkward expressions, the text is usually clear enough. Hopefully, the journal can assist the authors in improving the text further.

I now provide some minor comments for the authors.

The point the authors wish to make in Section 2.2, which is now very short, could be made more clearly, ie, the AI cosmologist was developed to for data analysis and paper writing and does not include observation planning and execution.

Greater clarity can be achieved by using more consistent terminology. For example, in section 4.1, the terms “Central Agent” and “Central Main Agent.” The extra word in the latter places a seed of doubt in the mind of the reader. If these two terms refer to the same thing, then the same words should be used.

In section 4.4, the authors note that they apply a real-bogus model to remove spurious detections. Can more details be provided?

The meaning of the text starting with “while the real bogus ...” in the caption to Fig. 5 is unclear.

The lack of a detection may also be due to the data being obtained before explosion (line 298)

Was the transient in Fig. 6 correctly identified by the system automatically, or was manual intervention needed (line 311)

Some terms may not be familiar to readers who are new to natural language models. For example, the idea of tokens (line

340).

What does “dynamically improve” (line 409) mean?

Line 436. The GOTTA prototype is not described before this point. Perhaps add some text pointing the reader to the Appendix where it is described.

What is RAG? Line 444.

Fig. 8 There are no hexagons in the figure.

Is NAOC defined somewhere? (line 924)

Reviewer #3

(Remarks to the Author)

I thank the authors for their thorough and constructive response to the initial round of reviews. The revised manuscript has improved significantly in both content and clarity.

The authors have addressed the major concerns raised in my previous review, particularly regarding:

1. Quantitative Evaluation: The revised manuscript now includes useful system performance metrics, such as execution success rates, scheduling efficiency, and function call statistics. While a full ablation study is understandably difficult, the added data provide a much clearer picture of the system's capabilities.
2. System Limitations and Failure Modes: The discussion of hardware and software constraints, as well as mitigation strategies, has been expanded and now provides a more balanced assessment of real-world challenges.
3. Language and Structure: The text has been substantially edited for clarity, and technical jargon has been better defined. Redundant or preliminary content has been moved to the appendix, improving the flow of the main manuscript.
4. Overstated Claims: The claim about discovery lag has been revised to avoid overstatement and better reflect the current capabilities of the system.

Overall, the manuscript presents an innovative application of agent-based AI in time-domain astronomy and offers a valuable prototype for future autonomous systems. While some areas could benefit from further polish by a language editor, the revisions are sufficient for publication.

Recommendation: Accept after minor editorial review.

Version 2:

Reviewer comments:

Reviewer #2

(Remarks to the Author)

The authors have submitted a revised version of the paper describing their work.

While there are still some awkward expressions, the text is clear enough.

Recommendation: Accept

Reviewer #1 (Remarks to the Author):

This work presents StarWhisper, an agent-based observation assistant for astronomy. It puts into practice many recent ideas regarding agentic AI applied to discovery and follow-up of nearby galaxy supernovae based on a network of small aperture telescopes. The main claim of this work is to establish a new paradigm of embodied AI systems to accelerate discoveries in astrophysics.

The authors are experiencing with new AI tools that will change how astronomy (and most natural sciences) are currently done, and the exact combination of tools that are being used is useful information for the community. However, almost no effort is done to show data and evidence about how this system works. Also, very little is done comparing current efforts that achieve much faster reaction times than those discussed in this work (a few minutes) without using AI agents.

To merit publication I believe that apart from reporting the workflow in detail, which is very useful, a lot more evidence about the performance of the system should be provided, e.g., execution time statistics, failure rates, fraction of cases that required human intervention, and, crucially, relevant metrics that show an improvement over non AI agent-based system (if one claims to accelerate discoveries, evidence should be provided that this is the case).

I tend to agree with the authors that the future of astronomy will look like what they have done, and I congratulate them for their work. However, the evidence is not there.

Response:

We sincerely appreciate the referee's thoughtful feedback. In response, we have significantly revised the manuscript (in green color) to include additional details on system performance, including success rates of function calls and comparisons with human operations (Section 5), as well as a discussion of system limitations (Section 6.1).

The success rate of function execution was evaluated by repeatedly issuing current query prompts to the SWT system and measuring the proportion of successful function calls. The overall success rate across all functions is approximately 70%. Notably, the core function—observation scheduling—achieves a perfect 100% success rate. Other functions, such as loading transient sources or input data and telescope control, demonstrate success rates between 60% and 70%. However, the function responsible for uploading observation plans to the telescope has a lower success rate (~30%). This is primarily due to network instability at Xinglong Observatory during the testing period. While the system successfully transmits the schedule, confirmation messages from the telescope computer are often not received due to timeout errors caused by slow connections. In practice, however, the schedules are typically executed correctly unless there are server-side issues, such as memory overflows. The server also hosts other pipelines (e.g., miniSiTian or GOTTA prototypes), which may lead to resource contention.

The total computational runtime across 7620 executions is 871.6 minutes, involving approximately 58.6 million tokens. This does not include observational time or data pipeline processing time, which will be detailed in a forthcoming publication. The X-OPSTEP image processing pipeline runs concurrently with observations and takes about 1–3 minutes per image.

Quantitative comparison with manual operations is challenging. We conducted a test comparing observation schedule generation: SWT demonstrates clear advantages in terms of time efficiency, galaxy coverage, and constraint handling. Human operators, however, offer greater flexibility—for example, switching between scheduled observations and monitoring a single galaxy throughout the night (a mode occasionally requested by the head of Xinglong Observatory).

Regarding human intervention, since the SWT system cannot directly interact with hardware, observers must manually verify telescope status in case of errors. Additionally, tasks such as dome control, flat image are still performed manually due to the lack of automated hardware. These issues are largely mitigated by the observers of miniSiTian, which operates under the

same dome.

Assessing the impact on discovery acceleration is complex. While it is difficult to quantify numerically, an automated system inherently speeds up scientific discovery, and agent-based systems like SWT further enhance this by integrating multiple tools into a unified framework. SWT provides a higher level of automation through its ability to coordinate workflows, abstract away low-level data handling, and offer modular APIs for future enhancements, such as integration with Model Context Protocol (MCP). Before SWT, the NGSS system could only manage 2-3 telescopes simultaneously, suggesting that our system has indeed accelerated observational capacity and discovery potential.

Lastly, the observational results are somewhat limited by the capabilities of the small-aperture telescopes, which constrain factors such as depth and resolution.

Reviewer #2 (Remarks to the Author):

Some of the terminology is unclear. Examples include: “a limiting observational personnel sequence” in the introduction, “industrial computer memory explosion,” in section 5 and “directional position deviation” in section 6. There are also awkward phrases, grammatical errors, incomplete sentences, typos (e.g., “amputator-level” which should presumably be “amateur-level”), and spelling mistakes. I recommend that the journal use the services of a language editor to help the authors express their ideas more clearly.

There is some jargon, for example, “AI agent workflow” and some less well-known acronyms are not defined, e.g. UUID. Not everyone will be familiar with UUID and in this particular case, the definition used in

Listing 1 (on p. 30) differs from what I had understood this acronym to mean. If one were to Google this acronym, then one would find Universally Unique Identifier.

The paper can be reduced significantly by removing some of the repetitive text and focussing on what has been achieved rather than what is planned.

Response:

We sincerely thank the reviewer for the thoughtful and constructive feedback, which has greatly improved the clarity and overall quality of our manuscript. In response, we have thoroughly revised the text (in green color) to address all concerns.

Regarding the use of technical terms and jargon—such as “AI agent workflow”—we acknowledge that some expressions may not be immediately clear to all readers. We have simplified or clarified such terminology throughout the manuscript to enhance readability and accessibility.

With respect to the acronym “UUID,” we appreciate the reviewer’s comment. While in our context it refers to a unique identifier generated by Python code to label individual processes, we now explicitly define it as “Universally Unique Identifier” upon first mention to avoid confusion, in line with its commonly accepted meaning.

In addition, we have moved content related to future plans and preliminary efforts involving the SiTian (GOTTA) project to the Appendix. This change helps streamline the main text and keeps the focus on the current achievements of the StarWhisper Telescope system.

Finally, we have carefully reviewed the entire manuscript to correct grammatical errors, spelling mistakes, and awkward phrasings. Unnecessary repetition has been removed to ensure a more concise and focused

presentation.

Once again, we are grateful to the reviewer for the helpful and detailed feedback, which has led to substantial improvements in both the language and structure of the paper.

Reviewer #3 (Remarks to the Author):

Main 1

•• Discovery lag is discussed in terms of anecdotal comparisons rather than systematic benchmarking. In fact, the claim of a median discovery lag of less than 12 hours for nearby transients appears only in the abstract, with no supporting data or detailed analysis in the main text.

Response:

We thank the referee for pointing out this issue. Upon reflection, we agree that the previous statement regarding a median discovery lag of less than 12 hours lacked sufficient supporting evidence and was potentially misleading.

What we intended to convey was a theoretical upper bound on discovery lag: for transients brighter than the limiting magnitude currently being monitored by our system, the discovery lag should be less than half of the observation cadence, which is approximately one day. However, due to the relatively small field of view and bright limiting magnitude of our telescopes, the effective revisit time for any given area is typically shorter than 12 hours for actively monitored regions.

Nonetheless, we acknowledge that this estimate was not rigorously validated with real data, nor was it representative of general performance across all observing conditions. As such, we have removed this statement from the revised manuscript to avoid misinterpretation.

Main 2

- The paper lacks a systematic evaluation of the system's impact: for example, the only quantitative comparison provided is that the assistant previously spent 2-3 hours per night preparing observations manually, time which is now saved through automation. However, this is presented without supporting metrics or an ablation study comparing performance with and without the agent-based architecture.

Response:

We thank the reviewer for this insightful comment. We acknowledge that a direct, systematic comparison between manual operations before and after the deployment of the agent-based architecture is challenging to conduct. This is largely due to the fact that prior to the implementation of the agent system, the NGSS telescope network was not fully operational — only one or two telescopes at Xinglong Observatory were used, and typically for very basic and infrequent tasks such as staring observations of M31.

In this context, the deployment of the agent-based architecture was in fact a foundational step that enabled the coordinated operation of the entire NGSS network. Without it, the current level of automation and scalability would not have been possible. The system has since served as a technical prototype and reference implementation for other projects such as GOTTA (the SiTian Pathfinder), as well as for integrating additional telescopes at Xinglong Observatory.

In Section 5, we present quantitative results on the system's execution success rates and performance metrics, including comparisons with manual operations in terms of time efficiency and observational coverage. Additionally, in Section 6.1, we provide a discussion of the system's

limitations and failure modes under real-world conditions.

Main 3

- Error rates from real/bogus classification are not reported.

Response:

We thank the referee for raising this point. In response, we have removed the relevant paragraph from the current manuscript, as the real/bogus classification component is not central to the core contributions of this work.

For completeness, we note that the classification module is based on a ResNet architecture enhanced with an attention mechanism—similar to the model used in the miniSiTian pipeline (see arXiv:2504.01608)—and achieved a validation accuracy of 99.12%. The model is trained on a NVIDIA RTX3090 GPU with 50 epochs. A detailed report on the real/bogus classification performance will be included in a forthcoming paper focused on the X-OPSTEP pipeline, which is currently under preparation. Additional tests, such as those on the ZTF dataset, are also planned but have not yet been completed.

The training dataset was constructed from manually vetted 64×64 pixel difference images, without access to the original template or science images. It includes fewer than 50 confirmed real transients. To compensate for the limited sample size, we applied data augmentation techniques to expand the real-transient set to approximately 2500 images.

The small number of real events is primarily due to the design of the NGSS observation strategy, which focuses on supernovae in nearby galaxies within 50 Mpc. As a result, image subtraction and transient detection are performed only in regions associated with known galaxies from the 50Mpc galaxy catalog, rather than across the full field using standard SExtractor-based source detection.

Main 4

- Known failure modes (e.g., hardware crashes, cold weather issues, software faults) are acknowledged, but no quantitative analysis or mitigation plan is presented.

Response:

We thank the reviewer for this comment. We have categorized failures into hardware- and software-related issues.

Hardware failures (e.g., camera malfunctions, cold weather effects) are beyond the control of the AI agent. Telescopes at Xinglong Observatory are maintained by SiTian Pathfinder technicians, while remote nodes often lack on-site support. We have standardized some components (e.g., ZWO ASI 6200 cameras) to improve reliability, though environmental issues still occur.

For software, crashes in NINA (mainly due to full disk storage) were a common issue. We have added a monitoring script to prevent such failures. While full quantitative analysis is difficult due to system heterogeneity, we have added a brief discussion of failure modes and mitigation strategies in Section 6.1 and 6.2.

Main 5

- Some references (e.g., “[2], [3] in prep.”) are placeholders and do not allow independent verification.

Response:

We thank the referee for this comment, we have deleted these references.

Main 6

Notably, Section 6.1 (Strengths and Challenges) is relatively brief compared to the detailed technical descriptions elsewhere in the manuscript (e.g., system design and future plans), and it does not sufficiently expand on limitations, trade-offs, or lessons learned from real-world operation.

Response:

We thank the referee for this comment. We have significantly revised the Section 6.1, adding more information about the limitations, and descriptions.

Main 7

- The English is often awkward, with grammatical errors and inconsistent phrasing throughout. Many sentences would benefit from language editing to improve readability and professionalism (see detailed list below).

Response:

We thank the referee about the detailed pdf file, especially about the part about the language edition. We revised the manuscript, and will present a new file in the appendix to response to some questions in the pdf file. Thank you again for you effort.

Main 8

- Some sections are overly dense, mixing implementation-level detail with conceptual discussion (e.g., system architecture and workflow descriptions), making it hard to follow the main thread.
- Figures are generally helpful but would benefit from more descriptive captions and consistent visual formatting.
- The length of the manuscript (including appendices) may exceed what is typical for Nature-style articles and could be better organized by moving technical implementation details to supplementary material.

Response:

We thank the referee for these three comments about the structure of manuscript, we have significantly revised the manuscript by putting many low correlated works to the appendix, and adding more supporting material. For example, the running log of SWT system to Github.

The PDF attachment.

We sincerely thank the referee for the valuable comments, especially those pointing out unclear language and descriptions. We have revised, added, and removed content throughout the manuscript to address these concerns, and we provide responses to the key issues below.

1.

"...can be halted through the Telescope Control workflow."

Question

Can you elaborate on how is this done? Does the Weather Monitoring agent send a "stop" command to the Telescope Control workflow? Can it send a "start" command too?

Response:

We thank the referee for this question. Yes it can, both of them, although the control of dome is not fully automated, so it sounds make no sense.

2.

"...we issue commands to the agent..."

Question

Is this a manual step or is this automated? If it is manual, why, when you automatically terminate the observations at astro. dawn?

Response:

In agent, it's a automated step. The dome should be open manually by the observers, though, before observation, the observer should take flat image, the dome and telescope cover will be opened.

3.

"...high-priority objects..."

Question

Are these objects observed, so they are going to be reobserved, or are they unobserved? Or a mix of both?

Response:

These are the mix of both. The high-priority objects come from potential transients, TNS reports, and manually assigned objects. A potential transient is observed and going to be reobserved. A TNS report may not be observed previously, as well as the assigned object.

4.

"...reassigned for the next day."

Question / Rword

The sentence is talking about a list that is generated based off the previous night's observations. So in that case, do you mean that these unobserved ones are reassigned to the current night, or do you actually mean they aren't reassigned until the next day?

Response:

Sorry for not express clearly. It means that when we found potential transients at day T, then they will appears in the day T+1.

5.

"...that output standardized data from the telescope..."

Question

Is this correct? The systems being monitored are not only coming from the telescopes, right? For instance the weather station, cloud map, allsky camera aren't from the telescope itself but from other supporting subsystems?

Response:

Thank you for pointing out this problem! The monitoring is come from other supporting subsystems, including the monitors for telescopes and the weather station.

6.

"...is an up-and-coming technique during observations and will help astronomers gain more information and inspire new insights."

Question / Reword

Is there a reason you are mentioning this new technique but not providing any additional information about it? Elaborate and explain how this technique relates to the topic of the paper. Also, it should be reworded as: "...up-and-coming technique used during observations that helps astronomers gain more information and inspire new insights into their area of science."

Response:

We thank the referee for pointing out this issue. We have done a lot of developing on automatic data analysis agent, which we believe holds substantial promise for enhancing observational astronomy. This agent is designed to process data immediately as it is collected, providing real-time insights akin to having a dedicated scientist on-site guiding the observations. Such capability ensures that valuable insights are promptly

identified and acted upon, potentially transforming observational strategies and outcomes.

An illustrative example of this approach can be seen in the AI cosmologist described in arXiv:2504.03424, where automated workflows replicate many manual processes with high fidelity. This not only streamlines operations but also fosters deeper scientific understanding by rapidly generating actionable insights.

6.

"If the problem persists or is more severe,..."

Question

How does the agent know that a problems severity has increased?

Response:

This is indeed part of our future plans. The determination of whether a problem's severity has increased depends on whether human intervention is required. This need for human intervention is assessed based on the maintenance logs, operational logs, and observation logs collected from the telescopes. These logs can be aligned with monitoring data using methods such as CLIP, and further use RAG to link this dataset with LLM. When the large model accesses this aligned dataset, it can determine the severity of the issue. If the problem requires human intervention, it is flagged as more severe. By integrating these logs and aligning them with monitoring data, our system will be able to accurately assess and respond to the severity of problems.

7.

"Custom plugins...to meet specifit needs."

Question / Rerword Can you elaborate by givning an example?

Response:

Sure. For example, autofocus, exposure calculator, moon angle, three point polar alignment.

8.

Questions

The paragraph is a bit confusing. Are you saying that your detection date lagged by 1-2 days from the TNS report, as shown in Table 3. What do you mean when you said you checked the data though? Does it mean that you observed the same targets but couldn't detect the transients as early as the TNS report? "...while our observing time is earlier than the discovery time reported." I think you mean to say that "...while our observing time is later than the discovery time reported.

Response:

We thank the referee for this good question. This question is mainly about the construction of data pipeline. In a time domain pipeline, we do the image subtraction and detection at once, the object detected will be sent to a internal alert. In our pipeline, x-opstep, the detection is mainly done by SExtractor after the subtraction process done by hotpants, with a 3-sigma detection. When the background is too noisy, the 3-sigma detection will cover too many detected sources since the sigma became larger. This is the reason why we have observed the target, but the pipeline didn't send an alert. So when an TNS target is reported, we often look back to our data to do force photometry to find out if this target is also discovered by our telescope. The discovery of the target in our telescope is earlier than the report at TNS, but not sent as an alert by the pipeline. The force photometry is done manually before, and now under developing, aiming to gather it into a new tool named Virtual GOTTA (something like the beneath image, also an AI tool). The virtual GOTTA will be soon applied to replace the agent reporting module in SWT since it include more information from other surveys.

Latest Candidates (5 sources)

AT 2025pxk

RA: 208.687317°

Dec: 0.117325°

Discovery Date: 2025-06-29
04:40:15.997

Discoverer: ZTF

Host Galaxy: CGCG 018-001 PED02

Host Type: G

Host Redshift: 0.0299

Xinglong Observability: Observability unavailable (missing astropy/astroplan)

Light Curve

▶ **ATLAS Forced Photometry (93 points)**

▶ **ZTF Observations (4 points)**

Dear Referee:

We sincerely thank you for your thoughtful, detailed, and constructive comments on our manuscript. Your insightful feedback has been invaluable in helping us improve the clarity, accuracy, and overall quality of the paper.

We have carefully considered each of your suggestions and have revised the manuscript accordingly. All changes have been made to enhance the presentation and accessibility of our work; the reversion is in green with a [] bracket.

We are truly grateful for your time, expertise, and the care you have taken in reviewing our work. Your comments reflect a deep understanding of the subject and have significantly clarified the manuscript.

Thank you again for your guidance and support.

Cunshi Wang

Referee Response

1.

In section 4.4, the authors note that they apply a real-bogus model to remove spurious detections. Can more details be provided?

Response: We thank the referee for raising this point. In response, we have updated the manuscript to include some more details about the real-bogus classification module.

The classification model is based on a ResNet architecture augmented with an attention layer—similar to the approach used in the current miniSiTian pipeline—and achieved a validation accuracy of 99.12%. Training was performed on an NVIDIA RTX 3090 GPU over 50 epochs.

For training, we constructed a dataset consisting of manually vetted 64×64 pixel difference images, without using the original template or science images. This dataset initially contained fewer than 50 confirmed real transients. To address the limited number of real transient examples, we applied extensive data augmentation

techniques to expand the real-transient sample to approximately 2,500 images. The augmentation methods include those used in the miniSiTian pipeline—such as flipping, rotation, scaling, and brightness adjustments—with the addition of a custom noise injection technique applied directly to the difference images to enhance robustness further.

A comprehensive evaluation of the real-bogus classifier’s performance, along with further architectural details, will be presented in an upcoming paper dedicated to the X-OPSTEP pipeline, which is currently in preparation.

2.

The meaning of the text starting with “while the real bogus …” in the caption to Fig. 5 is unclear.

Response: We thank the referee for pointing this out. The sentence beginning with “while the real-bogus…” in the caption of Figure 5 has been removed, as it was misleading.

To clarify: the input to the real-bogus classification module is a 64×64 pixel cutout centered on the candidate detection. However, the image shown in Figure 5 displays a larger field of view, encompassing the entire host galaxy. This full-frame image is saved for manual inspection and contextual analysis but is not used by the real-bogus classifier. We have revised the caption to better reflect this distinction and avoid confusion.

3.

Was the transient in Fig. 6 correctly identified by the system automatically, or was manual intervention needed (line 311)

Response:

We thank the referee for this insightful question.

The transient candidate shown in Figure 6 was initially detected automatically by our pipeline and was later reviewed and confirmed by a human observer. The extent of manual intervention in the validation process depends on the desired balance between detection efficiency and scientific reliability.

In principle, the system is capable of fully autonomous operation. However, current pipelines still face practical challenges—including telescope-specific characteristics, variable weather conditions, image artifacts, and algorithmic limitations. Even with a robust real-bogus classifier, a significant number of false positives persist, and some faint or morphologically complex transients may be missed.

To mitigate these issues, human inspectors currently play a critical and multifaceted role. This includes visual validation of candidates and triggering follow-up observations—such as those conducted with the 2.16m telescope. This human-in-the-loop approach remains essential for ensuring high-confidence transient identification and enabling reliable follow-up campaigns.

That said, manual inspection is labor-intensive. Although the NGSS survey and the GOTTA project (including its pathfinder and prototype telescopes) operate independently, they share the same team of personnel for visual inspection. This overlap places significant strain on human resources. As a result, NGSS candidates are assigned lower priority in the manual review queue and are primarily processed through automated channels, with only occasional manual checks performed when capacity allows.

We note that even with a real-bogus classifier, some false positives remain, and the system cannot yet achieve perfect reliability. The current values hold that humans are still more accurate than models. Therefore, while automation is our primary mode of operation, occasional human oversight remains valuable for validation.

4.

Some terms may not be familiar to readers who are new to natural language models. For example, the idea of tokens (line 340).

Response: We thank the referee for this helpful suggestion. To improve accessibility

for readers who may be less familiar with NLP terminology, we have added brief explanations of key terms in the manuscript.

Specifically, we now include a short description of *tokens*—the basic units of text (such as words or subwords) processed by language models—and provide context for their usage in our workflow. For transparency, we also report the estimated cost associated with the number of tokens used, based on standard cloud service pricing.

Additionally, we have added concise introductions to *RAG* (Retrieval-Augmented Generation, a method that enhances model responses by retrieving relevant information from external sources) and *MCP* (Model Context Protocol, a framework for integrating tools and context into model interactions).

5.

What does “dynamically improve” (line 409) mean?

Response: We thank the referee for pointing out the ambiguity in this phrasing. The term “dynamically improve” has been removed to avoid confusion. We clarify that the intended meaning is that improvements to the system are expected to be implemented consistently and incrementally throughout the observation. The core meaning remains unchanged, and the revised text now more clearly reflects this ongoing, systematic refinement process.

6.

Line 436. The GOTTA prototype is not described before this point. Perhaps add some text pointing the reader to the Appendix where it is described.

Response: We thank the referee for this suggestion. We have added a brief introductory phrase at the first mention of the GOTTA prototype to provide context and direct the reader to Appendix 2, where a more detailed description of the telescope and its role in the project is provided.

7.

What is RAG? Line 444.

Response: We thank the referee for this comment. RAG stands for *Retrieval-Augmented Generation*, a framework that enhances the responses of large language models by retrieving and incorporating relevant information from external knowledge sources. As noted in the manuscript, RAG is introduced in Section 6.1. We have now added an explicit reference to this section at the first mention of RAG to ensure readers can easily locate the explanation.

8.

The point the authors wish to make in Section 2.2, which is now very short, could be made more clearly, ie, the AI cosmologist was developed to for data analysis and paper writing and does not include observation planning and execution.

Greater clarity can be achieved by using more consistent terminology. For example, in section 4.1, the terms “Central Agent” and “Central Main Agent.” The extra word in the latter places a seed of doubt in he mind of the reader. If these two terms refer to the same thing, then the same words should be used.

Fig. 8 There are no hexagons in the figure.

Is NAOC defined somewhere? (line 924)

Response: We thank the referee for these helpful comments. We have carefully revised the manuscript. We added some descriptions in Section 2.2 to clarify this. The terminology has been revised carefully. We also removed the descriptions about hexagons in Figure 8, and clarified the NAOC as the National Astronomical Observatories, Chinese Academy of Sciences.